# Optimal Design of a Sensor Network for Guided Wave-Based Structural Health Monitoring Using Acoustically Coupled Optical Fibers

**DOI:** 10.3390/s24196354

**Published:** 2024-09-30

**Authors:** Rohan Soman, Jee Myung Kim, Alex Boyer, Kara Peters

**Affiliations:** 1Institute of Fluid Flow Machinery, Polish Academy of Sciences, Fiszera 14, 80-231 Gdansk, Poland; 2Department of Mechanical and Aerospace Engineering, North Carolina State University, Campus Box 7910, Raleigh, NC 27695, USA; jkim79@ncsu.edu (J.M.K.); amboyer2@ncsu.edu (A.B.); kjpeters@ncsu.edu (K.P.)

**Keywords:** guided waves, fiber Bragg grating (FBG) sensors, acoustic coupling, multi-objective optimization

## Abstract

Guided waves (GW) allow fast inspection of a large area and hence have received great interest from the structural health monitoring (SHM) community. Fiber Bragg grating (FBG) sensors offer several advantages but their use has been limited for the GW sensing due to its limited sensitivity. FBG sensors in the edge-filtering configuration have overcome this issue with sensitivity and there is a renewed interest in their use. Unfortunately, the FBG sensors and the equipment needed for interrogation is quite expensive, and hence their number is restricted. In the previous work by the authors, the number and location of the actuators was optimized for developing a SHM system with a single sensor and multiple actuators. But through the use of the phenomenon of acoustic coupling, multiple locations on the structure may be interrogated with a single FBG sensor. As a result, a sensor network with multiple sensing locations and a few actuators is feasible and cost effective. This paper develops a two-step methodology for the optimization of an actuator–sensor network harnessing the acoustic coupling ability of FBG sensors. In the first stage, the actuator–sensor network is optimized based on the application demands (coverage with at least three actuator–sensor pairs) and the cost of the instrumentation. In the second stage, an acoustic coupler network is designed to ensure high-fidelity measurements with minimal interference from other bond locations (overlap of measurements) as well as interference from features in the acoustically coupled circuit (fiber end, coupler, etc.). The non-sorting genetic algorithm (NSGA-II) is implemented for finding the optimal solution for both problems. The analytical implementation of the cost function is validated experimentally. The results show that the optimization does indeed have the potential to improve the quality of SHM while reducing the instrumentation costs significantly.

## 1. Introduction

The global structural health monitoring (SHM) market is expected to be USD 3.8 billion in 2027 [1]. The field of SHM enables reliable, sustainable maintenance protocols for efficient performance and prolonged service life of structures. SHM has been adopted in various sectors, including civil, offshore, and aerospace structures. The SHM techniques are classified based on different damage sensitive features used, such as vibration [2], strain [3], guided wave propagation (GW) [4], electromechanical impedance [5], etc.

Of these techniques, guided wave monitoring is commonly used for monitoring of large thin-walled structures. Several actuation and sensing transducers are used for the generation and detection of the guided waves. Piezoelectric materials such as lead zirconate titanate (PZT), macro-fiber composites and magneto-restrictive materials are the most commonly used for actuation [6]. These actuation devices have been paired with optical fiber-based sensors such as Fabry–Perot interferometers or fiber Bragg grating (FBG) sensors [7,8] or non-contact techniques such as scanning laser Doppler vibrometry (SLDV) [9]. Optical fiber-based sensors are advantageous because FBGs are insensitive to electrical and magnetic fields, small in size and light in weight.

The edge-filtering approach is commonly applied for detection of low-magnitude guided waves [10]. A tunable laser is tuned to the midpoint of the rising or falling edge of the FBG-reflected spectrum. As a result, a small shift in the reflectivity spectrum yields a significantly large change in the reflected optical power. Moreover, the FBG sensitivity can be further enhanced by employing a remote bonding configuration [11]. In the remote bonding configuration, the optical fiber is bonded to the structure away from the FBG location. The guided waves propagating in the structure are coupled from the structure into the optical fiber as guided longitudinal modes ($L_{01}$), which are sensed by the FBG further along the optical fiber [12]. As the FBG is not constrained by the adhesive bond to the structure, remote bonding under certain conditions leads to enhanced sensitivity of the sensors to guided waves compared to directly bonded FBGs.

Recently, researchers showed that optical fibers can also be acoustically coupled to transfer longitudinal waves travelling along one optical fiber to another. The coupling can be achieved by fusion-splicing two optical fiber cores to each other [13,14]. This process requires precision manufacturing and is nearly impossible to achieve in situ for an in-service structure. More recently, the authors reported that the acoustic coupling may be achieved by simply adhesively bonding the two optical fibers [15,16]. This simple adhesion can be paired with the remote bonding technique to design an SHM adaptive network to interrogate multiple locations on a structure with only a single FBG sensor. A single FBG sensor reduces the instrumentation complexity without compromising the quality of monitoring a large area structure. Earlier, the authors provided a proof of concept for the acoustic coupling concept for damage localization [17]. The technique has also been used to improve the coverage of an existing network in the presence of new damage [18]. Soman et al. [17] used a 3 × 1 acoustic coupler to detect guided waves at three locations in a structure using only a single FBG sensor. The three branches of optical fiber (one with FBG and two ordinary polyimide-coated single-mode fibers) were coupled using cyanoacrylate glue. The measurements were temporally separated by employing optical fibers with different lengths. In order to avoid inaccurate identifications of the wave packets due to overlaps between the arrival times of the reflections from each branch and optical fiber ends, three critical lengths had to be chosen deliberately: the length of each branch, measured from the structure to the coupler; the length between the coupler and the FBG sensor; and the length between the FBG and the free end of the optical fiber (since reflections are present).

These demonstrations show that signals can be multiplexed to a single fiber (and therefore a single FBG detector), but they do not answer the question as to whether using this multiplexing approach increases the resultant signal amplitude and therefore improves the damage identification. Additionally, a second question is as follows: If the signal amplitude is increased, which coupler configuration optimizes the amplitude? The advantages to multiplexing signals are that they reduce the complexity of the FBG interrogation system. However, the costs to multiplexing are the signal loss at each coupler. The loss is proportional to the input mode amplitude, but also depends on the coupler configuration (1 × 2, 2 × 2, 3 × 1, etc.). For example, the losses in the two configurations shown in Figure 1 are potentially different, even though they achieve the same combination of three original Lamb wave signals.

This paper serves as an extension to the authors’ previous research efforts in Soman et al. [17,19] to answer these questions by developing an actuator–sensor system including a design of the optical coupler configuration itself. We propose a two-step multi-objective optimization to achieve this goal. In the first stage, the locations of the actuators and the sensors are optimized to maximize the spatial coverage of the sensing network. Then, the acoustic coupler dimensions are optimized in the second stage to maximize the final output signal. The rest of the paper is organized as follows: Section 2 reports preliminary experiments to measure the energy transfer for different coupler configurations which serve as the inputs and constraints for the optimization problem. Section 3 explains the definition of the optimization problem and its implementation. Section 4 presents the results of the optimization. Section 5 discusses a limited validation of our proposed optimization. Finally, conclusions related to validity and accuracy are discussed in the last section.

## 2. Measurement of Coupler Outputs and Losses

To define the optimization problem, we first need to know how much of the input signal is transferred to the different branches of the acoustic coupler and how much is lost for all possible coupler configurations. Based on initial studies, the number of incoming branches was restricted to a maximum of three, as a higher number of branches reduces the signal-to-noise ratio (SNR) and also increases the fiber lengths significantly. Previous work by the authors focused on 2 × 2 couplers to replicate the common geometry of fused optical fiber couplers [17,18]. But in this paper, we want to optimize the coupling by using the minimum number of branches needed. Therefore, the optimization problem was reduced to two coupler choices, the 2 × 1 and 3 × 1 cases shown in Figure 1.

### 2.1. Experimental Methods

The proportion of $L_{01}$ mode energy coupled into each branch in a 2 × 1 coupler and a 3 × 1 coupler were first determined experimentally. Standard, 125 μm diameter optical fibers with a thin polyimide coating were used for all the optical fibers. Figure 2a shows the experimental setup for the 3 × 1 coupler. The setup for the 2 × 1 coupler was the same, with only one fewer fiber into the coupler. One of input branches to the coupler was adhesively bonded to a a thin aluminum plate, radially aligned with the PZT. The $S_{0}$ mode generated in the plate by the PZT was converted into a $L_{01}$ mode in the input optical fiber at the adhesive bond location. Figure 2a shows the case where the middle branch is coupled to the plate for the 3 × 1 coupler.

Details on the coupler fabrication by adhesively bonding a portion of the two or three fibers are given in [16]. The acoustic coupler length is 3.175 mm. This length is chosen based on previously fabricated couplers that performed well [17]. The lengths of the optical fibres were chosen such that the direct arrivals from each of the bonds do not overlap with the reflections from the coupler or the end of the fiber. The details of the factors affecting the coupler are briefly discussed by the authors in [17].

The input $L_{01}$ mode was measured using the FBG located on the fiber bonded to the plate. For the case in Figure 2b, this is FBG 2. Since the input fibers were bonded and removed from the plate several times, measuring the input $L_{01}$ mode normalized out the effects of any differences in the adhesive bonds. After the $L_{01}$ signal passed through the acoustic coupler, the output signal was measured with the FBG in the output branch (FBG 4 in Figure 2c) and the reflections back into each of the input branches was measured with the FBGs in each of those optical fibers (FBGs 1–3 in Figure 2c). The FBGs in each of the optical fibers were chosen randomly and were between 1532 nm and 1564 nm. The measurements were normalized with the slope of the each FBG to ensure that the measurements from each of the FBGs are comparable. The detailed procedure for the process may be found in [11]. Once the measurements were complete, the adhesive bond bonding the optical fiber to plate was removed, another branch of the coupler was bonded to the plate in the same manner and the measurements were repeated.

The $L_{01}$ mode signal amplitude through each coupler branch was measured for 5 separate fabricated couplers to verify the repeatability of the fabrication process and signal collection. The signal energy was calculated for each case, proportional to the square of the waveform amplitude. The energy lost at the coupler was also calculated as the total energy input minus the energy output after the coupler. Finally, all energy values were normalized to the input energy for that coupler. From these measurements, three energy values were calculated: the sum of the energy reflected back through the input 2 or 3 branches of the coupler; the energy output through the 1 output branch; and the energy loss which is the input energy minus the energy reflected and output. The experiment was repeated for each input branch of the coupler to test the symmetry of the coupler outputs.

### 2.2. Coupler Energy Distribution Results

The average values over the 5 couplers for the 2 × 1 and 3 × 1 couplers are plotted in Figure 3. For both coupler configurations, the output ranges from approximately 15 to 25% of the input signal. For both configurations, the losses and back reflections are high. The reflection can be basically considered as a loss as well, as the energy is not coupled into the output branch. By far, the majority of the energy reflected was back into the fiber with the input signal for all cases. These losses are consistent with previous experiments by the authors [17,18] and are primarily due to the manual fabrication method and the fact that the ultrasonic energy is widely distributed through the optical fiber; therefore, a significant portion leaks into the environment at the coupler. The inherent damping properties of the polymer adhesive may also contribute to the loss. Improving the output of the adhesive bond coupler is important for practical applications; however, it is the subject of other work, while the focus of this article is the optimization method.

The encouraging result for this work is that the output is consistent (within the standard deviation) for the 2 × 1 and 3 × 1 configurations for each of the input branches, meaning that the output is symmetric. For the 2 × 1 coupler, the output values were 23.2% ± 4.4% and 23.7% ± 7.6% using Branches 1 and 2 as the input fiber, respectively. For the 3 × 1 coupler, the output values were 16.2% ± 2.6%, 15.5% ± 2.3%, and 18.4% ± 2.8% using Branches 1, 2 and 3 as the input fiber, respectively. This symmetry means that the energy coupled from any branch produces approximately the same output, so the optimization results are not extremely sensitive to the order of the connections in the sensor network. For practical applications, this is important since sometimes physical constraints restrict the order in which the fibers are arranged.

The average output energy for the 3 × 1 coupler did drop by 29% compared to the 2 × 1 coupler (from 23.5% to 16.6%), The variation between the different branches increased slightly as well in the 3 × 1 coupler. As the cross-section of the coupler increased and was likely slightly distorted for the 3 × 1 configuration, this decrease in output is reasonable. A 4 × 1 coupler was fabricated as well; however, the output energy further decreased for this case, resulting in a low SNR and so was not considered in this paper.

The energy throughput, reflection and loss for these two coupler configurations are used as input to the optimization problem in the next section.

## 3. Optimization Problem Definition

The design of an optimal SHM system using acoustically coupled FBG sensors has two components: (a) the optimization of the actuator–sensor (AS) network based on the application demands and the limitations of the sensors and (b) optimal design of the acoustic coupler which include the factors mentioned earlier such as branch lengths, distance between FBG and the coupler and distance of the FBG from the fiber end.

These two components may be dealt with together as a single optimization problem or may be split into two separate optimization problems. In the current study, the problems are dealt with separately. The reasons are two-fold: the application demands from the AS network are of higher importance to the quality of the SHM achieved and as such have precedence over the design of the acoustic coupler. Secondly, one of the foreseen applications of the acoustic coupling is to retro-fit new sensors to an existing SHM network so that the SHM network may be extended using acoustically coupled FBG sensors to allow improved SHM at a location of possible damage. In such a case, the location of the sensors is dictated by engineering judgment and the design of the coupler should occur independently.

### 3.1. Optimization of AS Network

The aim of this stage is the optimization of the AS system to maximize the performance of the SHM system which uses an acoustic coupler to interrogate multiple locations with a single FBG sensor on a simple 0.5 m × 0.5 m aluminum plate. The scope of the problem was restricted to the optimization of the number and location of actuators, and the number, location and orientation of the bond locations which are the interrogation points.

The optical fiber shows directional sensitivity. It has been shown by Soman et al. [20] that in a remote bonding configuration, the coupled waves have a cosine proportionality to the wave coupled into the fiber. Hence, in the cases of angles exceeding 75 degrees, the $A_{0}$ wave is not sensed reliably.

The objectives identified for the application are coverage of the structure with at least 3 AS pairs (cov3) and the cost of the deployed AS network. The coverage with at least three AS pairs ensures the identification of the unique damage location using the intersection of the ellipse approach. The technique for the damage localization based on ellipse approach is discussed in [21,22].

The cov3 values were determined by using a stored coverage approach. This has been shown to be computationally more efficient. The coverage based on time of flight for all the possible actuator–sensor combinations for all bond orientations was calculated and stored in the form of a matrix. These matrices were then recalled and superimposed to obtain the coverage matrix for a given actuator–sensor network. This approach imposes a non-physical constraint on the optimization problem. The actuator and sensor locations are restricted to only integer encoded positions as shown in Figure 4. In reality, they can take any position on the structure. Similarly, a constraint on the angular orientation of the sensor is imposed to compute the coverage before the optimization process. This, although not ideal, is necessary to limit the problem size and the computational costs. The velocity for the computation of the direct arrivals and the reflections was taken as 5390 m/s for the $S_{0}$ wave and 2450 m/s for the $A_{0}$ wave [17]. It was assumed that the $A_{0}$ wave would be used for damage localization as the smaller wavelength makes it better for detection of small cracks [23]. The limit on the time of flight was determined based on the minimum time of arrival of the first edge reflection. This limitation simplifies the signal processing and improves the damage localization performance of the network. This parameter may be changed depending on the damage localization algorithm employed as proposed in [24].

The cost of the AS network was calculated based on the number of sensors and actuators. The cost of an FBG sensor and the apparatus for the interrogation is much more expensive than the actuation system for the PZT sensor. Hence, the the cost of each actuator was taken as 10 units, while that of FBG sensors was taken as 1000 units. The assumed costs are in keeping with the cost of the sensor and actuator systems. A key difference from the previous work presented in [25] is that in the earlier case, acoustic coupling was not used, hence each bond location required an independent FBG sensor and an interrogation system. But in this case, through acoustic coupling, up to three bond locations can be interrogated by a single FBG. Hence, the cost was calculated by the following equation:(1)$$cost=10\times N_{A}+1000\times ceil\left(\frac{N_{S}}{3}\right)$$
where $N_{A}$ is the number of actuators, function ‘ceil’ rounds to the minimum integer greater than the fraction, and $N_{S}$ is the number of sensors. The restriction of three bond connections to a single FBG was determined based on experiments in Section 2.

The joint optimization of the number of actuators and sensors (bonds), their location and the orientation was performed. For a multi-objective optimization, the two objectives may be combined together into one after assigning relative weights through scalarization, but the relative importance of the metrics may not be a priori known. Hence, a true multi-objective optimization is necessary. For the application at hand, the non-sorting genetic algorithm (NSGA-II) was implemented due to its ease of use and ability to handle both integer and real valued variables. The flowchart of the NSGA-II is shown in Figure 5. The authors highlight that there are several other optimization algorithms that are capable of performing true multiobjective optimization such as multiobjective particle swarm optimization [26], Pareto Envelope-based Selection Algorithm [27], etc. But as the focus of the paper is on the design of the acoustic coupler and not the optimization algorithm, the NSGA-II was employed. The AS network was encoded as a 19-gene-long chromosome. Each gene was an integer. The first two genes were for the number of actuators and sensors, respectively. The next five genes were assigned to the actuators. If fewer than five actuators were used, the remaining genes were assigned a value of zero. The next six genes were allocated for the locations of sensors. The actuator and sensor genes take any integer value from 0 to 81. The next six genes were assigned to the orientation of the bond. As the plate is isotropic and symmetric, the orientation could take integer values from 0 to 135. A two-point crossover was implemented. More details about the NSGA-II and its implementation may be found in [25]. The implemented NSGA-II was shown to have excellent convergence so the selection strategy, crossover operator and mutation operator were kept identical to the one in [25]. These parameters need to be tuned to ensure that indeed the global optimum is achieved. But as the focus of the study was more to design the acoustic coupler than the optimization algorithm, that is identified as a topic of future work.

### 3.2. Optimal Design of Acoustic Coupler

Once the locations of the actuators and the sensors are determined, they need to be integrated in the acoustically coupled network. The factors determining the performance of the coupler include high SNR for the measurements, the ability to separate the measurements at the different locations in time and the ability to differentiate the signals captured from the structure and the reflections from the artefacts in the optical fiber network (coupler, fiber ends, splices, etc.). The FBG sensor in the remote configuration is more sensitive to the $L_{01}$ wave. Also, the attenuation coefficient of the $F_{11}$ wave (0.5 $m^{-1}$) is much higher than the coefficient for the $L_{01}$ wave (0.19 $m^{-1}$). So only the $L_{01}$ wave and its reflections are considered. The $L_{01}$ wave velocity is taken as 5110 m/s [11]. The proportion of energy coupled in each branch of the coupler is taken as presented in Section 2. The reflection coefficient of the wave from the edge and the bond location is taken as 20% based on the experience of the authors.

The three objectives for the optimization are (a) the overlap between the measurements from the three bonds, (b) the overlap between the reflections from the coupler and the end of the fiber and (c) the total length of the fiber. The overlap between measurements relates directly to the ability to attribute the measurements to the different bonds. The time windows for the arrival of the direct wave and the reflection from the edges of the plate is determined for each actuator and bond location. The separation of these time windows is determined as the metric for the optimization. The goal is to remove the overlap (or at least minimize). The second metric directly relates to the ability to differentiate between signals coming from the structure and the signals arising from the reflections in the optical fibres. This metric too is determined based on the time windows of the wave arrivals and the arrival of the reflections from the coupler as well as from the ends of the fiber. The third metric is the total length of the fiber used for the network. Although the fiber is relatively cheap, the long fibers are more difficult to manage and are more prone to breakage as a result as metric of the price. To ensure ease of handling, this objective is included.

The NSGA-II is then implemented with these objectives to determine the branch length for the optimal coupler. The chromosome length for the NSGA-II is determined based on the optimal sensor placement obtained from the first stage. The branch length for each of the sensors is one of the decision variable. The length between the coupler and the FBG and the length of fiber between the FBG and the fiber end are the other variables. So for a three-sensor network, the chromosome length is five. The fiber lengths can take any real values, so in order to limit the artificial constraints on the encoding, a real valued NSGA-II is used. During the deployment of the sensors the fiber length needs to be rounded off to the least count of measurement device. The minor rounding-off errors does not affect the optimal results significantly but helps in the fast convergence of the solution.

## 4. Results and Discussion

The section discusses the results obtained for the two-step optimization process. The results are presented and discussed in the following subsections.

### 4.1. Optimization of AS Network

The cost function identified was the coverage with three AS pairs. The Pareto front (Figure 6) presents the optimal coverage achieved for different costs of the AS network.

As the cost is dependent on the number of FBG sensors, there is a large jump on the axis corresponding to the cost. Another interesting observation is the NSGA-II favors the use of actuators more than the sensors. This is obvious from the fact that a sensor placement with cost 2020 (six sensors and two actuators) or 2010 (six sensors and one actuators) is not seen in the Pareto front.

In our case, only one tunable laser and an amplified photodetector are available. So an arrangement with three sensors needs to be chosen for deployment. The coverage plot for the optimal configuration with three sensors and three actuators is shown in Figure 7.

### 4.2. Optimization of Acoustic Coupler

Once the locations of the actuators and sensor are known, the acoustic coupler design can be optimized. The real valued NSGA-II was implemented. The Pareto front is shown in Figure 8. In order to show the effectiveness of the NSGA-II implemented, the Pareto front was compared with the Pareto front obtained by using multi-objective particle swarm optimization (MOPSO). The details of the MOPSO implementation and the algorithm can be found in [28,29]. The comparative plots are shown in Figure 9. As can be seen, the Pareto fronts are reasonably in agreement but not overlapping. This is due to the fact that both the algorithms show convergence but are not reaching the global optimum within the number of iterations or generation specified in the optimization. This could be improved, but at the cost of computational effort and time. Also, as the hyper-parameters were not tuned, the computational performance of the two algorithms could not be compared with each other. The tuning of the hyper-parameters, however, is an interesting area of research considered beyond the scope of this work and the interested readers are referred to [30].

As can be seen in the Pareto front, as the length of the optical fiber increases, the overlap in the signals as well as the overlap with the reflections tends to decrease, which is expected. The arrangement with no overlap for the reflections as well as for the direct arrivals has a total length of 9.1 m.

### 4.3. Validation

The aim of this paper is to outline the methodology for the two-step optimization of the design of the AS network for damage localization and the design of the acoustic coupler to obtain high fidelity information from the AS system. The validation of the methodology can be carried out by ascertaining that the different objective functions implemented for the optimization have physical significance and can be validated experimentally. In this section, the validation of the physical meaning of these objective functions is carried out on the aluminium plate. This plate experiment was not designed for the optimized coupler, but we originally designed for the characterization of the 3 × 1 coupler. But this study clearly indicates that the predictions of the identified cost functions for this realized setup are correct and hence there is physical significance to the implemented cost functions. The actuator and sensor locations were determined by the needs for the characterization of the coupler. But the analytical prediction of the coverage parameter agrees very well with the experiments. The damage scenarios simulated for the plate in the region covered by the network are easily located by the algorithm. The actuator was located at (0.19, 0.197) and (0.293, 0.445). The bond locations and orientations were determined not by the optimization but by the needs of the experiment for the 3 × 1 coupler characterization. The setup is shown in Figure 10. The damage was simulated with additional mass in the form of magnets (disc shaped with a diameter of 2 cm and weight of 25 g). This is a commonly applied strategy for introducing reversible damage in SHM studies [31].

The coverage obtained analytically for the setup is shown in Figure 11. The experimental studies were conducted with simulated damage in the region shown in Figure 11. The methodology for the damage localization including the signal processing is explained in detail in [17]. Some exemplary results are shown in Figure 12 and Figure 13. All seven scenarios are in the coverage range and were indeed localized with the AS network.

This shows that the implementation of the coverage objective of the first stage of the optimization is indeed physically valid and predicts the damage localization performance of the AS network satisfactorily.

In the second stage, the acoustic coupler implemented is checked for accuracy with the analytical time of arrival. Figure 14 shows the experimental results and the analytical prediction for the arrival.

The time of arrival is in excellent agreement with the analytical predictions. Due to the uncertainty in the bonding of the fiber to the plate as well as the acoustic coupler realized, the amplitude of the coupled wave cannot be compared directly.

## 5. Conclusions

The use of acoustic coupling has the potential to enhance the quality of SHM possible with minimal increase in the cost. This is possible as the acoustic coupling allows interrogation of multiple locations with the single FBG sensor. The ability has a potential to reduce instrumentation costs by over 50%. This paper provides a framework for the design of an SHM system harnessing acoustic coupling using a two-step approach. In the first stage, the AS network placement is optimized based on the application demands for the SHM. A multi-objective optimization problem is formulated with the cost of the equipment and the coverage of the AS network as the objectives. An integer-based NSGA-II is implemented. The integer NSGA-II imposes some additional constraints on the placement of the actuators and the sensors but restricts the problem size significantly allowing a faster convergence to the optimal solution. The computational load is significantly reduced due to the integer encoding and hence the imposition of additional constraints is considered acceptable. Once the locations of the AS network are determined, the sensor network making use of acoustic coupling is developed. Again, a multi-objective optimization problem is formulated with three objectives corresponding to the time separation of the measurements from the different bonds, the separation of the measurements from the optical features (reflections) and the total length of the fiber.

A limited validation is provided to show the physical significance of the objective functions defined as well as the accurate analytical implementation. The analytical coverage prediction is validated with experimental studies. Similarly, the time of arrival of the direct waves from the three bonds is shown to be in agreement with the experiments. This shows that the implementation is indeed physically relevant and accurately implemented. The method has the potential to improve the quality of SHM significantly while reducing the cost of instrumentation for the SHM system. The two-step process also allows the implementation of the design of the acoustic coupling directly for retrofitting of structures with monitoring.

Going ahead, the application of this methodology for designing and retrofitting sensor networks for real scenarios including crack detection will be carried out. The acoustic coupling is a very powerful tool, but in order to have clear distinction between the coupled waves at different branches, the edge reflections from the boundaries of the the structure need to be limited. As a result, the application of this technique is better aligned for large plate-like structures. The authors also acknowledge that the implementation of the locations of the bonds as integers imposes an additional constraint but is necessary to limit the problem size. In order to overcome this limitation, more work on the optimization algorithm and its efficient implementation and parameter tuning is needed.

## Figures and Tables

**Figure 1 sensors-24-06354-f001:**
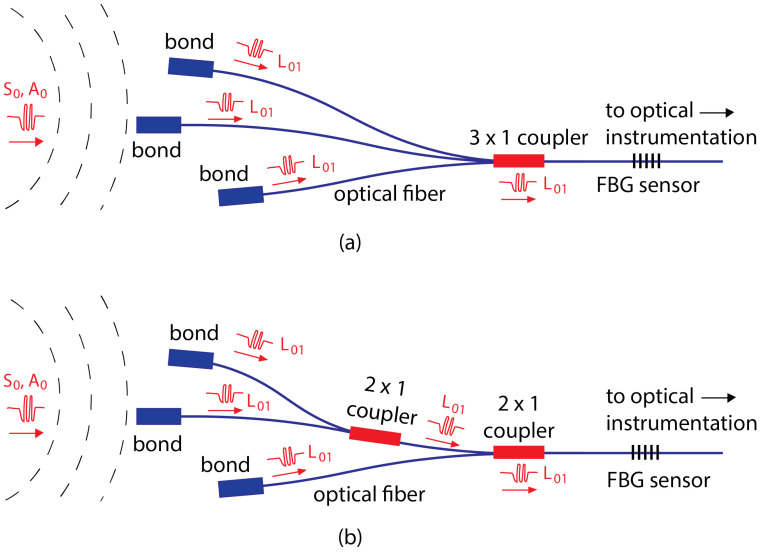
Coupling of three Lamb wave signals into a single FBG detector through (**a**) one 3 × 1 coupler and (**b**) two 2 × 1 couplers. Ultrasonic waves shown in red.

**Figure 2 sensors-24-06354-f002:**
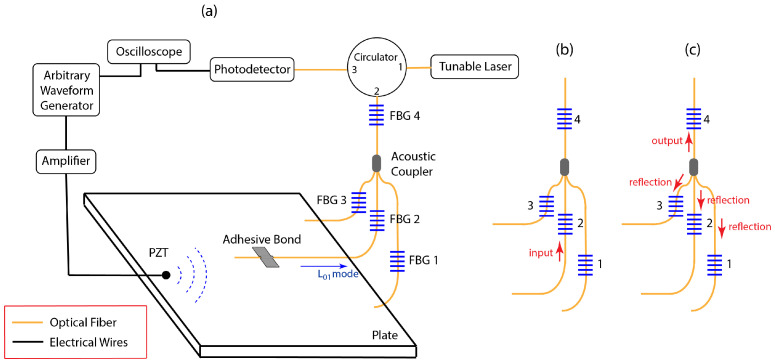
(**a**) Experimental setup for characterizing the different bonds. (**b**) Wave travelling to the acoustic coupler. (**c**) Energy distribution in all the branches after the coupler.

**Figure 3 sensors-24-06354-f003:**
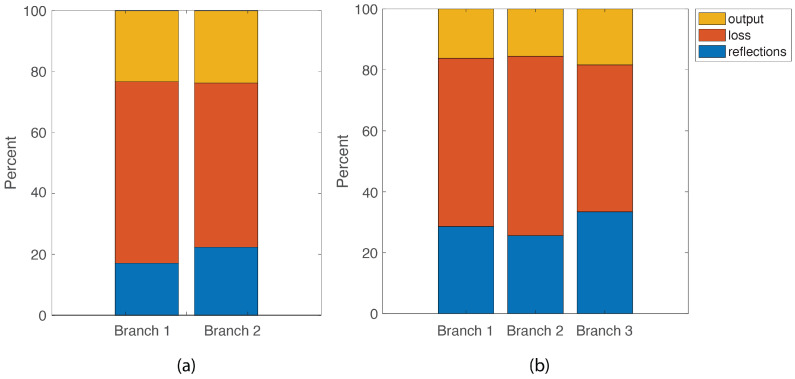
Percent of energy reflected, lost and output from a (**a**) 2 × 1 couplerm (**b**) a 3 × 1 coupler. Horizontal axis refers to the branch into which the $L_{01}$ is input.

**Figure 4 sensors-24-06354-f004:**
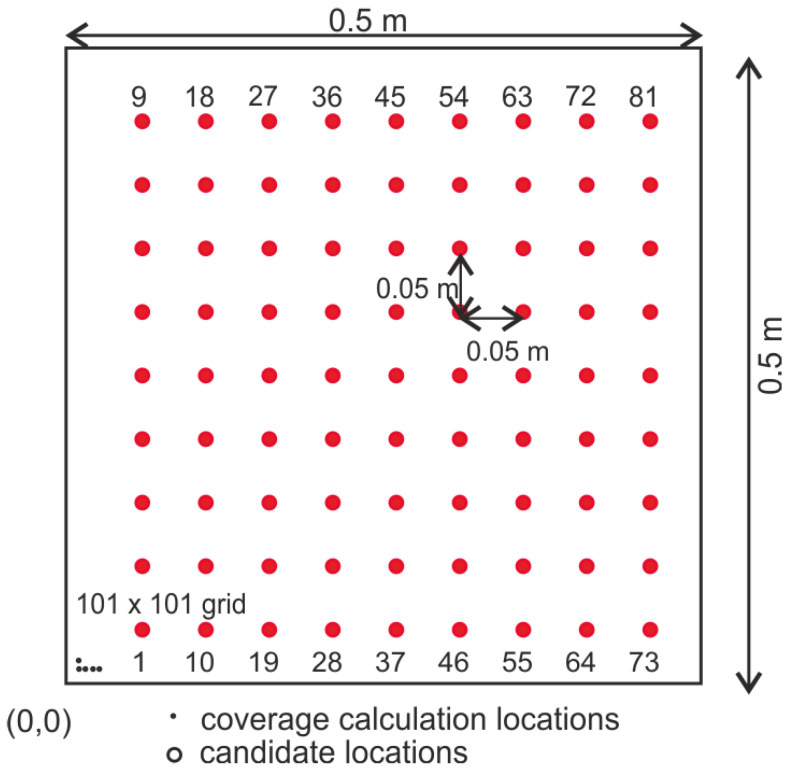
Candidate locations and discretization for coverage calculations.

**Figure 5 sensors-24-06354-f005:**
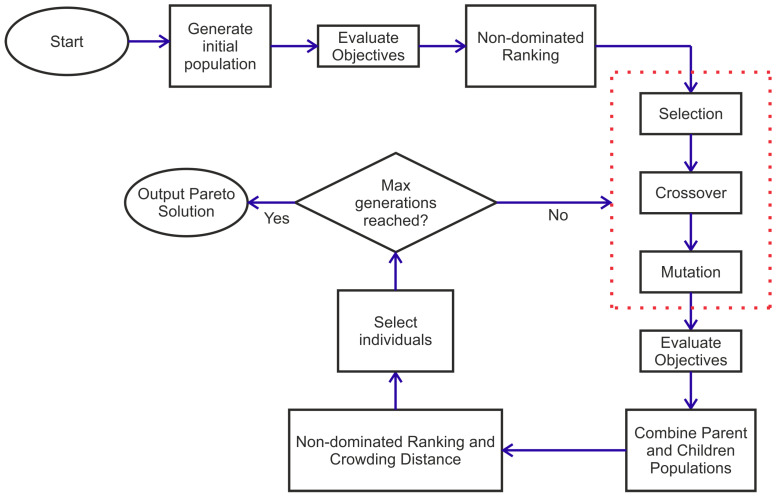
Flowchart for the NSGA-II based on [25].

**Figure 6 sensors-24-06354-f006:**
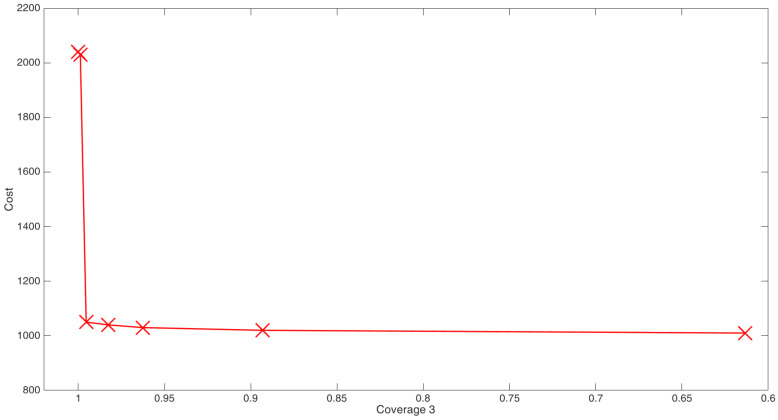
Pareto front for the optimization of AS network.

**Figure 7 sensors-24-06354-f007:**
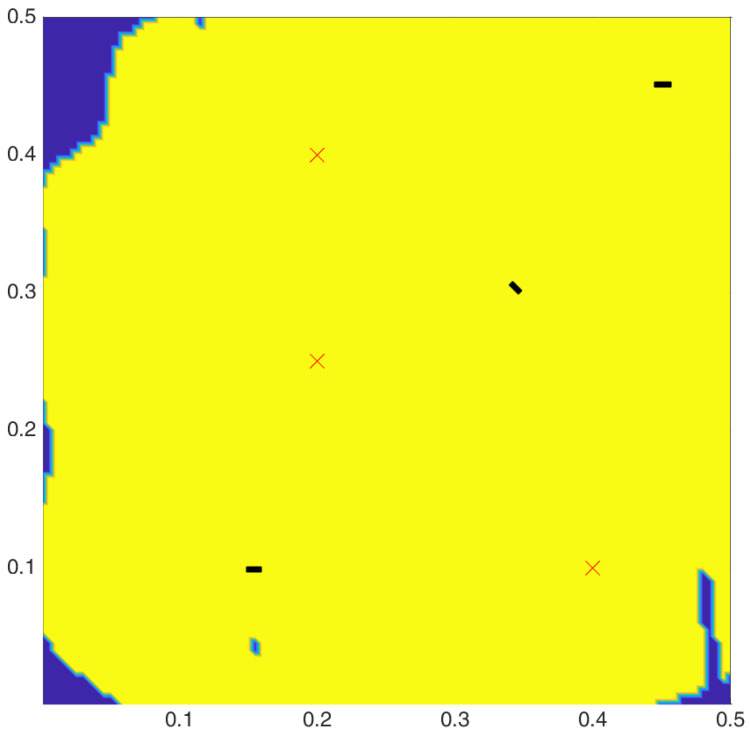
Surface plot for AS network (yellow shows covered areas, ‘X’ are the actuators, ‘rectangles’ are sensors showing orientation).

**Figure 8 sensors-24-06354-f008:**
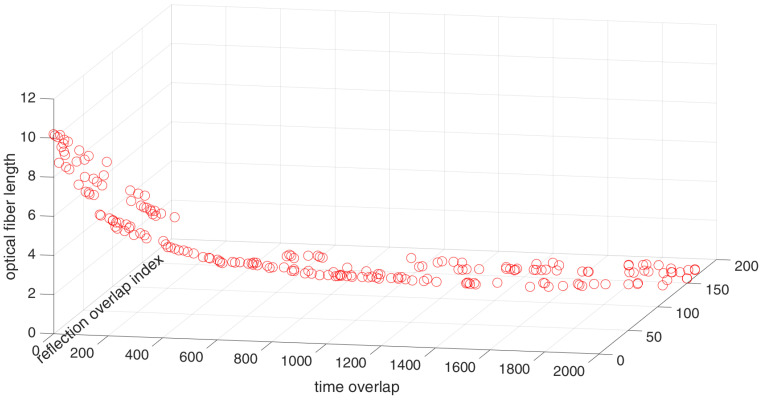
Pareto front for the optimization of acoustic coupler.

**Figure 9 sensors-24-06354-f009:**
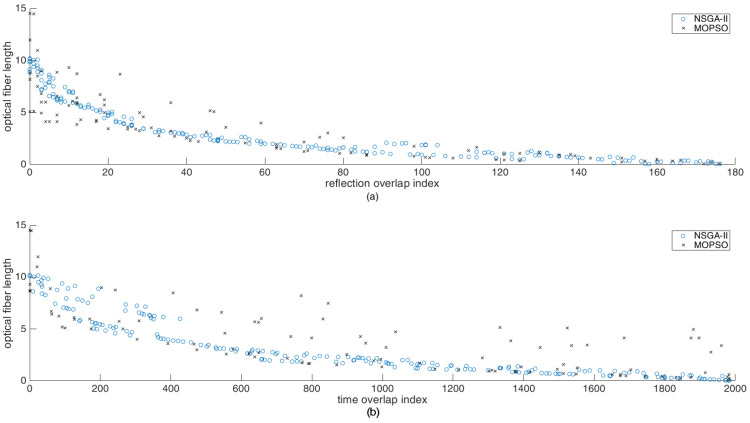
Pareto front for the optimization of acoustic coupler for better visualization in 2D.

**Figure 10 sensors-24-06354-f010:**
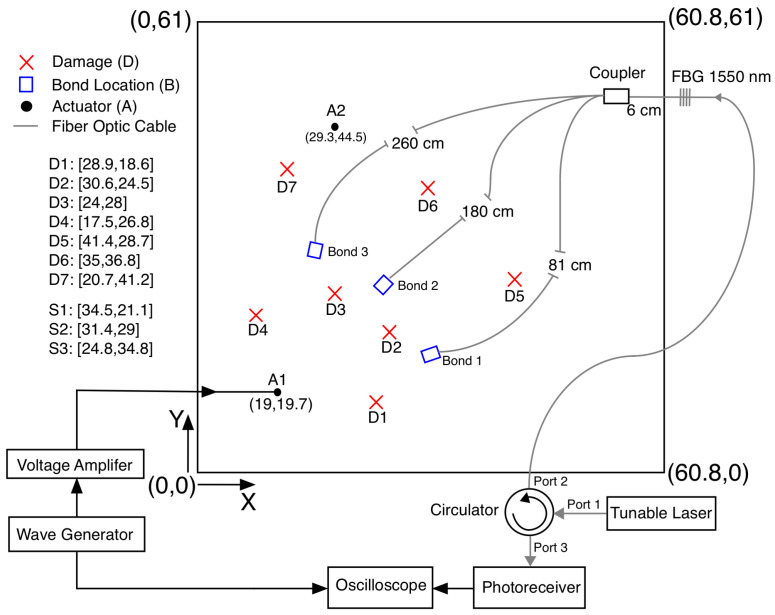
Experimental setup for validation (coordinates and dimensions in cm; reprinted with permission from [17]).

**Figure 11 sensors-24-06354-f011:**
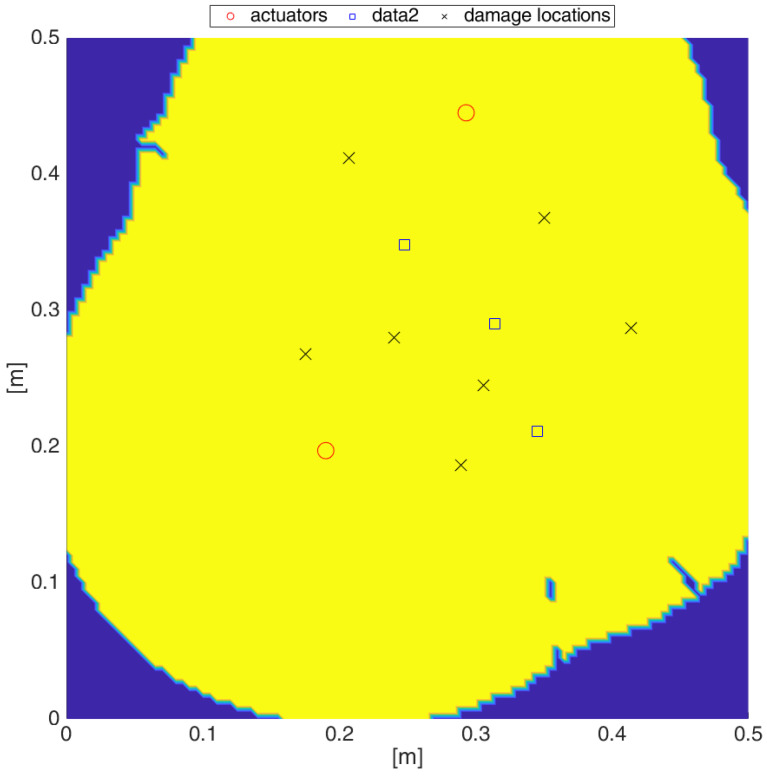
Coverage for the AS network (yellow shows covered areas, ‘o’ are the actuators, ‘rectangles’ are sensors, ‘x’ are the damage locations simulated).

**Figure 12 sensors-24-06354-f012:**
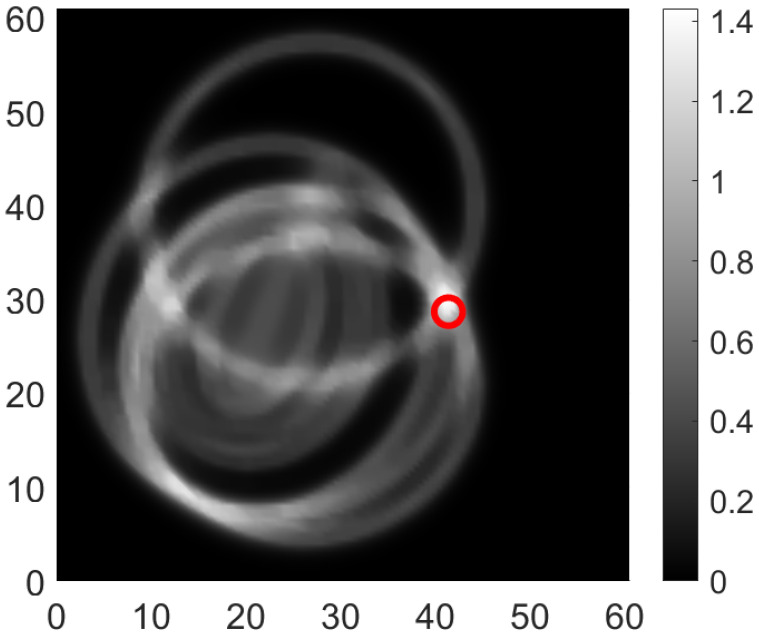
Localization of damage for Case D5 [17].

**Figure 13 sensors-24-06354-f013:**
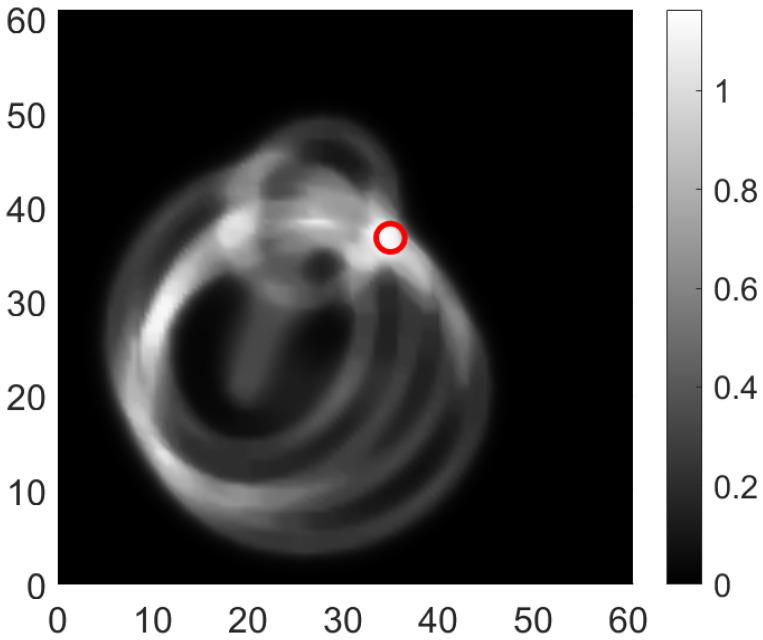
Localization of damage for Case D6 [17].

**Figure 14 sensors-24-06354-f014:**
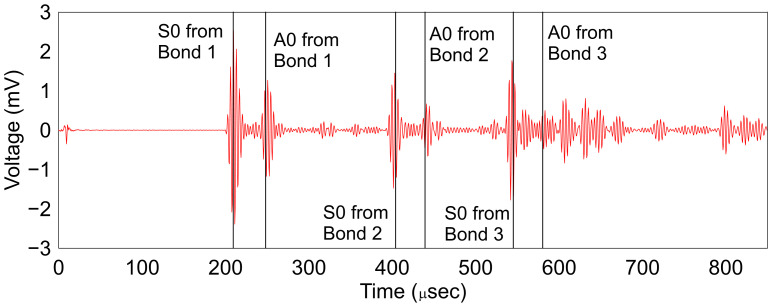
Agreement of analytical and experimental time of arrival.

## Data Availability

Data used in Section 2 available on request due to restrictions. Data from Section 3 onwards is published with DOI: 10.5281/zenodo.13580492.

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
