# Peer review of "Optimal Design of a Sensor Network for Guided Wave-Based Structural Health Monitoring Using Acoustically Coupled Optical Fibers"

_sensors, 2024, doi:10.3390/s24196354_

Round 1

Reviewer 1 Report

Comments and Suggestions for Authors

       The authors reported an optimal method of a FBG network for structural health monitoring applications based on guided waves and acoustic coupler. The optimal method is a two-step methodology, which aims to decrease the cost and ensure high fidelity measurements. The results will be interest to the community. Here are some questions.

1. The caption of Figure 2 is wrong.

2. Please give more explanation about acoustic coupler.

3. Why is the number of FBG in Figure 9 only 1 but it is 4 in Figure 2?

4. Line 304 claimed that there are 5 actuators in Figure 7, but in fact there are only 3 actuators.

5. The positions of actuators and sensors in optimization stage in Figure 7 were different from those in validation stage in Figure 9. Please explain this difference.

Comments on the Quality of English Language

None.

Reviewer 2 Report

Comments and Suggestions for Authors

This work continues a series of papers, prepared by the same author research team and focused on development of structural health monitoring (SHM) system by using acoustically coupled optical fibers. Here authors present approach and methodology for the design of SHM system, that includes procedure of optimization for actuator-sensor (AS) placement over the tested plate with following design of an acoustic coupler network to minimize distortions, occurring due to interference, reflections, loss etc. effects in optical fibers / fiber optic connections / acoustically coupled circuits. Authors successfully experimentally validated proposed method and confirmed its efficiency, that provides improvement of designed SHM system under reducing the instrumentation costs.

The paper is well prepared, it corresponds to the journal scope. It is suitable for publication in Sensors after some minor corrections / suggestions and answering on following questions / comments:

1.              Which type of optical fiber was used in the setup? Singlemode (SMF) or multimode optical fiber (MMF)? It seems, acoustic coupling is provided better for large core optical fibers.

2.              Following up on the previous question, what is the Bragg wavelength of recorded FBGs? Are the same for coupler branches (1…3 and 4) or not?

3.              How did authors precisely control and fixed length of branches? As well as track for branch placement over tested plate?

4.              Did the authors make attempts to denoise oscillograms by filter or any other digital signal processing tools / tricks?

5.                  Fig. 9 contains misprint: “Damge”.

Reviewer 3 Report

Comments and Suggestions for Authors

(1) The paper contains some grammatical errors and irregularities, such as the use of a in line 22, the omission of where in line 236, the absence of a figure number in line 327, and the missing punctuation at the end of line 331. The authors need to carefully proofread the entire manuscript.

(2) Lines 37-39: The authors mention that structural health monitoring (SHM) has been applied in many fields, but the classification of SHM techniques is unclear. The reviewer suggests establishing a clear standard to categorize existing technologies  to facilitate readers general understanding of SHM.

(3) Lines 243-245: The authors employ the non-sorting genetic algorithm (NSGA-II) for optimization, but the rationale for this choice is overly simplified. Each method has advantages and disadvantages. For instance, PSO offers the benefit of fully utilizing inertia weight (as seen in Damage Identification of Steel Bridge Based on Data Augmentation and Adaptive Optimization Neural Network, P6-8). The authors are advised to compare their chosen method with commonly used optimization algorithms to justify their decision.

(4) In Figure 5, based on the reviewers experience, genetic algorithms (GA) are known for their slow convergence and tendency to get trapped in local optima. The authors have not indicated whether any improvements have been made to the method. They are encouraged to elaborate on the selection strategy, crossover operator, mutation operator, and the use of maximum convergence generations to terminate iteration. Additionally, they could consider defining a termination criterion based on the change in the objective function falling below a certain threshold.

(5) The font size in Figures 6, 11, and 12 is too large. Please adjust it.

(6) In Figure 8, how can the decreasing trend of the reflections overlap index be discerned? If possible, consider adding another figure to illustrate this trend more clearly.

(7) In Figure 9, the authors should provide detailed information about the specific locations, types, and extents of the seven damage scenarios.

(8) In Figure 9 of the paper, two actuators are employed. If this is the case, how is the sensor network optimization process implemented? The authors are requested to further elaborate on this aspect.

(9) The methods described in the paper are applicable to plate-type structures, but it is unclear whether these methods are suitable for more common issues such as cracks in beam-type structures. The author is requested to address this issue in the section of conclusions.

Comments on the Quality of English Language

The paper contains some grammatical errors and irregularities, such as the use of a in line 22, the omission of where in line 236, the absence of a figure number in line 327, and the missing punctuation at the end of line 331. The authors need to carefully proofread the entire manuscript.

Round 2

Reviewer 3 Report

Comments and Suggestions for Authors

The third comment wasn't well addressed, the authors are advised to compare their chosen method with commonly used optimization algorithms, such as PSO,  to justify their decision.

Author Response

The third comment wasn't well addressed, the authors are advised to compare their chosen method with commonly used optimization algorithms, such as PSO,  to justify their decision.

Thank you for the comment. A MOPSO implementation was customized for the application at hand. After 200 iterations and without any hyper parameter tuning the results are similar to NSGA-II. The comparison is now added to figure 9, with a small discussion and appropriate references.

The computational performance cannot be compared as the hyper parameters were not tuned for this application. Also the 2 pareto fronts are reasonably in agreement which indeed shows that the NSGA-II implemented was valid.